# PARP1 Regulates Circular RNA Biogenesis though Control of Transcriptional Dynamics

**DOI:** 10.3390/cells12081160

**Published:** 2023-04-14

**Authors:** Rebekah Eleazer, Kalpani De Silva, Kalina Andreeva, Zoe Jenkins, Nour Osmani, Eric C. Rouchka, Yvonne Fondufe-Mittendorf

**Affiliations:** 1Department of Molecular and Cellular Biochemistry, University of Kentucky, Lexington, KY 40536, USA; rebekah.eleazer@uky.edu (R.E.); zoedjenkins@gmail.com (Z.J.); 2Department of Epigenetics, Van Andel Institute, Grand Rapids, MI 49503, USA; nour.osmani@vai.org; 3Department of Neuroscience Training, University of Louisville, Louisville, KY 40292, USA; kalpani.desilva@louisville.edu (K.D.S.); kalina.andreeva@louisville.edu (K.A.); 4Kentucky IDeA Networks of Biomedical Research Excellence Bioinformatics Core, University of Louisville, Louisville, KY 40292, USA; eric.rouchka@louisville.edu; 5Department of Biochemistry and Molecular Genetics, University of Louisville, Louisville, KY 40202, USA

**Keywords:** circRNAs, transcription elongation, poly (ADP-ribose) polymerase, splicing, backsplicing, gene architecture, RNA polymerase II elongation and pausing

## Abstract

Circular RNAs (circRNAs) are a recently discovered class of RNAs derived from protein-coding genes that have important biological and pathological roles. They are formed through backsplicing during co-transcriptional alternative splicing; however, the unified mechanism that accounts for backsplicing decisions remains unclear. Factors that regulate the transcriptional timing and spatial organization of pre-mRNA, including RNAPII kinetics, the availability of splicing factors, and features of gene architecture, have been shown to influence backsplicing decisions. Poly (ADP-ribose) polymerase I (PARP1) regulates alternative splicing through both its presence on chromatin as well as its PARylation activity. However, no studies have investigated PARP1’s possible role in regulating circRNA biogenesis. Here, we hypothesized that PARP1’s role in splicing extends to circRNA biogenesis. Our results identify many unique circRNAs in PARP1 depletion and PARylation-inhibited conditions compared to the wild type. We found that while all genes producing circRNAs share gene architecture features common to circRNA host genes, genes producing circRNAs in PARP1 knockdown conditions had longer upstream introns than downstream introns, whereas flanking introns in wild type host genes were symmetrical. Interestingly, we found that the behavior of PARP1 in regulating RNAPII pausing is distinct between these two classes of host genes. We conclude that the PARP1 pausing of RNAPII works within the context of gene architecture to regulate transcriptional kinetics, and therefore circRNA biogenesis. Furthermore, this regulation of PARP1 within host genes acts to fine tune their transcriptional output with implications in gene function.

## 1. Introduction

Poly (ADP-ribose) polymerase 1 (PARP1) is a highly abundant nuclear enzyme that uses NAD+ to synthesize and catalyze the addition of ADP-ribose polymers to target molecules, namely, histones and itself. The modification of histones with negatively charged PAR molecules disrupts the electrostatic interactions between histones and DNA to open the chromatin structure. This phenomenon is well understood in the context of DNA damage and repair, where PARP1 acts as a guardian or surveyor of the genome [1]. PARP1 is dynamically distributed on chromatin where it constantly scans the DNA [2]. The recognition of DNA damage by PARP1, for which it has high affinity, results in acute PARylation [3,4]. The acute PARylation of histones local to sites of DNA damage is critical for both the swift recruitment of DNA repair enzymes and for the maintenance of the open chromatin structure required for their access to DNA [5]. While the importance of PARP1 in DNA damage response is well understood, it is becoming increasingly apparent that both PARP1’s presence on chromatin as well as its PARylation activity play a more constitutive role in gene expression under normal cell conditions.

The effect of PARP1 and its PARylation activity on gene expression is pleiotropic, regulating RNA biogenesis at multiple levels [6]. PARP1 can affect the quantity of transcripts produced from a gene through its control of transcriptional initiation, promoter-proximal pause and release, and rate of RNAPII during elongation [7,8]. At the same time, PARP1 can also affect the quality (or types) of transcripts produced by a gene at the level of splicing. PARP1 exerts control of splicing through its regulation of RNAPII kinetics during elongation as well as its ability to recruit and modulate the behavior of splicing factors through PARylation. Our previous studies using PARP1 knockdown and PARP1-inhibited cells demonstrate that both PARP1 presence on chromatin and its PARylation have measurable impacts on gene expression, splicing decisions, and RNAPII kinetics throughout gene bodies [8,9,10,11]. However, the ability to predict whether PARP1 promotes exon inclusion or exon skipping depends on additional chromatin context at a given gene locus. Clearly, much more work is required to resolve our understanding of PARP1’s regulation of splicing, and one form of alternative splicing for which PARP1’s influence remains completely uninvestigated is backsplicing.

Backsplicing occurs when a downstream splice site is connected to an upstream splice site that circularizes the transcript to form a circular RNA (circRNA) [12,13]. The widespread abundance and biological relevance of circRNA have recently come to light [14,15]. However, factors that mediate the biogenesis of circRNA are still being investigated. We hypothesize that PARP1’s regulation of alternative splicing could extend to the biogenesis of circRNA. Our studies and those of others showed a role of PARP1 in transcription initiation and elongation [7,9,10], features important for circRNA biogenesis. Another important factor critical for circularization is the binding of specific splicing factors to the flanking intronic sequences. PARP1 has been shown to PARylate splicing factors to activate them [6]. We therefore carried out a comprehensive study of PARP1 and the generation of circRNAs. We describe the circRNAs generated in the presence and absence of PARP1 and show the possible functions of these circRNAs. Finally, we analyze the function of PARP1 binding in regulating RNAPII elongation along introns and exons and show that PARP1 creates a transcriptional environment that aids in making specific circRNAs. 

## 2. Materials and Methods

### 2.1. S2 Cell Culture and siRNA-Mediated Knockdown and PARylation Inhibition

All experiments were performed using Drosophila melanogaster S2 cells (obtained from Thermo Fisher Scientific, Waltham, MA, USA). The S2 cells were grown in Schneider’s Drosophila media (Life Technologies, Austin, TX, USA) supplemented with 10% heat-inactivated fetal bovine serum (Sigma-Aldrich, St. Louis, MO, USA, 68178) and 100 U/mL penicillin and 100 µg/mL streptomycin at 25 °C. All experimental samples and controls were matched in terms of growth time and cell density. The knockdown of PARP1 was performed as described in Matveeva et al. [9]. Western blot analysis (Figure 1A and Appendix A) was used to confirm PARP1 depletion. The S2 cells were treated with 10 µM PJ34 (Sigma-Aldrich, St. Louis, MO, USA, 68178) for 16 h to inhibit PARylation activity.

### 2.2. S2 Cell Protein and Total RNA Extract Preparation

The S2 cells were pelleted, washed, and resuspended in 0.5 mL RIPA buffer supplemented with 1 mM PMSF (phenylmethylsulfonyl fluoride, Sigma-Aldrich, St. Louis, MO, USA, 68178, #10837091001), 1X protease inhibitor cocktail (EpiGentek, Farmingdale, NY, #R-1101), 10 mM 3-MBZ (3-methoxybenzylamine, Sigma-Aldrich, St. Louis, MO, USA, 68178, #159891), and 0.5 mM BZA (benzylamine, Sigma-Aldrich, St. Louis, MO, USA, 68178, #185701). The cell suspension was incubated on ice for 30 min, gently vortexed, sonicated for 6 cycles (10 s on/10 s off) (Bioruptor300, Diagenode, Denville, NJ, USA), and spun down at 13,000 rpm for 10 min. The supernatant was used for Western blot analyses. The total RNA was extracted from the washed and pelleted cells using the Quick-RNA Miniprep Kit (Cat. #R1054) as per the manufacturer’s protocol (Zymo Research, Irvine, CA, USA, 92614) and eluted in RNase-free water. The total RNA was submitted (Novogene, Sacramento, CA, USA, 95826) for ribosomal-depleted paired-end RNA sequencing.

### 2.3. Western Blots

Protein lysates were prepared in 1X SDS loading buffer, loaded on 10% SDS-PAGE gel, and electrophoresed at 150 V for 45 min. The samples were transferred to PVDF membrane (Thermo Scientific, Rockford, IL, USA) and subsequently incubated with primary antibodies for PARP1 and β-actin to demonstrate PARP1 knockdown. The Western blot-based detection was performed using alkaline phosphatase-coupled secondary antibodies (Sigma-Aldrich, St. Louis, MO, USA, 68178) with Amersham ECF substrate for visualization (GE Healthcare, Waukesha, WI, USA), and images were obtained using a Typhoon FLA 9500 (GE Healthcare, Piccataway, NJ, USA). ImageQuant TL software version 10.2 was used to quantify the protein signals.

### 2.4. Antibodies for Western Blot Analysis

The primary antibodies used were PARP1 C terminal, rabbit polyclonal (#39561, Active Motif, Carlsbad, CA, USA), PAR rabbit polyclonal (#4336-BPC-100, Trevigen, Gaithersburg, MD, USA), and Actin, mouse monoclonal (MA1-744, ThermoFisher Scientific, Waltham, MA, USA, 02451). The secondary antibodies used were both anti-rabbit (#A3687) and anti-mouse (#A3562) IgG (whole molecule) alkaline phosphatase antibodies (Sigma-Aldrich, St. Louis, MO, USA, 68178).

#### 2.4.1. circRNA Detection

The paired-end raw sequencing files were downloaded in FastQ [16] format from Novogene onto the KY INBRE server. The data included in this paper came from three sample groups: PARP KD, PARPi, and WT. Each group consisted of three replicates. The FastQC (version 0.10.1) toolkit [17] was used to examine the quality of the sequencing runs. Trimming was not necessary since all the reads were of high quality with Phred scores above 28. Next, the circRNAs were detected using the seekCRIT tool version 1.0.0.b [18].

#### 2.4.2. circRNA Validation

For each circRNA, two sets of primers were designed: (1) divergent primers to amplify across the backsplice junction, and (2) convergent primers to amplify a linear region within the circRNA. As exons appear in gDNA in a linear fashion, divergent primers should not amplify with this template (Appendix A). After amplification, the PCR amplified samples were run on 1% agarose gel, where a single band of the expected size was detected (Appendix A). These bands were isolated, purified, and sequenced using Sanger sequencing. These sequencing results confirmed the presence of the head-to-tail backsplicing of these circRNAs (Appendix A).

#### 2.4.3. Bulk RNA-Seq Analysis

The concatenated FastQ sequence files were directly aligned to the *Drosophila melanogaster* reference genome assembly (BDGP6) using STAR (version 2.6) [19], generating alignment files in bam format [19]. The number of successfully aligned raw reads for each of the samples ranged from ~100 to 126 million. The raw read counts were obtained from the STAR-aligned bam format files using HTSeq version 0.10.0 [17]. Then, the raw counts were normalized using the relative log expression (RLE) method and then filtered to exclude genes with fewer than 10 counts across the samples. Differential expression analysis was performed using DESeq2 [20]. The significant differentially expressed genes were extracted at the q value cutoff of 0.05. The R package clusterProfiler [21] was used to identify enriched Gene Ontology biological processes and KEGG pathways for sets of differentially expressed genes in each comparison. 

#### 2.4.4. Alternative Splicing Analysis

Two separate methods involving the tools DEXSeq2 [22] and rMATS [23] were used for alternative splicing detection. The first analysis was run using DEXSeq2 in a custom R script. DEXSeq2 breaks up the novel splice patterns into individual exons and reports differential usage on an exon level. The significant differentially expressed exons were extracted at the q value cutoff of 0.05. The second analysis was run using rMATS, which measures skipped exons (SE), alternative 5′ donor splice sites (A5SS), alternative 3′ acceptor splice sites (A3SS), mutually exclusive exons (MXE), and retained introns (RI) based on exon junction reads. The number of significant events (q < 0.05) for each of the three comparisons was reported.

#### 2.4.5. circRNA Function, Host Gene Architecture, and Transcription Analysis

The most frequent miRNA targets in each experimental circRNA group were identified from miRTarBase (Release 9.0) [24] relative to human miRNA seeds. The target functional annotation was performed using clusterProfiler. 

The host gene lengths, circRNA internal exon/intron lengths, flanking intron lengths in each condition (WT, PARPi, and KD), and other Drosophila gene lengths were analyzed using custom Python scripts. The introns flanking the 5′ and 3′ backsplice sites were examined for RBP motifs using rMAPS [25]. The G-quadruplexes in the host genes and other Drosophila genes were identified using quadparser (version 2.0) [26]. The G-quadruplex densities, nucleotide compositions, and the RNAP II pausing across the host genes in each experimental condition were calculated using custom Python scripts. 

#### 2.4.6. GEO Data Deposition

The RNA-seq data from these experiments are found in GEO # GSE225445. The PARP1-mediated RNA pausing data came from our previous studies on nascent RNA-seq ([10] and GEO GSE118266).

## 3. Results

### 3.1. Genome-Wide Identification of PARP1-Mediated circRNAs in S2 Drosophila Cells

We previously showed that PARP1 influences alternative splicing [9], yet the role of PARP1 in circRNA biogenesis has not yet been determined. We used S2 Drosophila cell lines to test whether the physical presence of PARP1 and/or its enzymatic activity alters the presence of circRNAs. Drosophila is an important model organism as it contains only one isoform of PARP1, negating the impact of redundancy that might be expected in humans with more than 18 PARPs. We knocked down PARP1 (KD) using siRNA (Figure 1A) and inhibited its PARylation activity (PARPi) by treating cells with PJ34, as previously performed [8,11]. Three independent ribosomal-depleted paired-end RNA-seq experiments, without RNase R digestion, were performed, generating ~100 to 126 million read pairs per sample. We used the computational pipeline SeekCRIT [18], which considers both linear and circular junctions, to identify not only the possible circRNAs, but also to measure changes in backsplicing relative to flanking linear splicing events (Figure 1B). We detected 284 unique backsplice junctions (364 total) categorized as either intronic (ciRNA) or exonic (circRNA) across all samples. We identified 70 unique intronic circRNAs (26 in WT, 21 in PARPi, and 23 in KD), as well as 214 unique exonic circRNAs (67 in WT, 87 in PARPi, and 60 in KD). The exonic circRNAs were further subdivided into single exonic circRNAs and multi-exonic circRNAs (Figure 1C). 

Exonic circRNAs are formed via a co-transcriptional pathway, while intronic circRNAs are regulated via post-transcriptional pathways [27,28]. Since PARP1 mediates co-transcriptional splicing [9,10,11], we focused the downstream analysis on these exonic circRNAs. The 284 unique circRNAs we detected are derived from 116 protein-coding genes (Figure 1C), suggesting that at least some host genes produced multiple circRNA isoforms. (Appendix A shows the complete list). On the other hand, while a significant number of detected circRNAs were unique among the experimental conditions, there was an overlap of the presence of some circRNAs (Figure 1D) and their host genes (Figure 1E) between the treatment groups. We concluded that circRNA is indeed present in our PARP1 model.

**Figure 1 cells-12-01160-f001:**
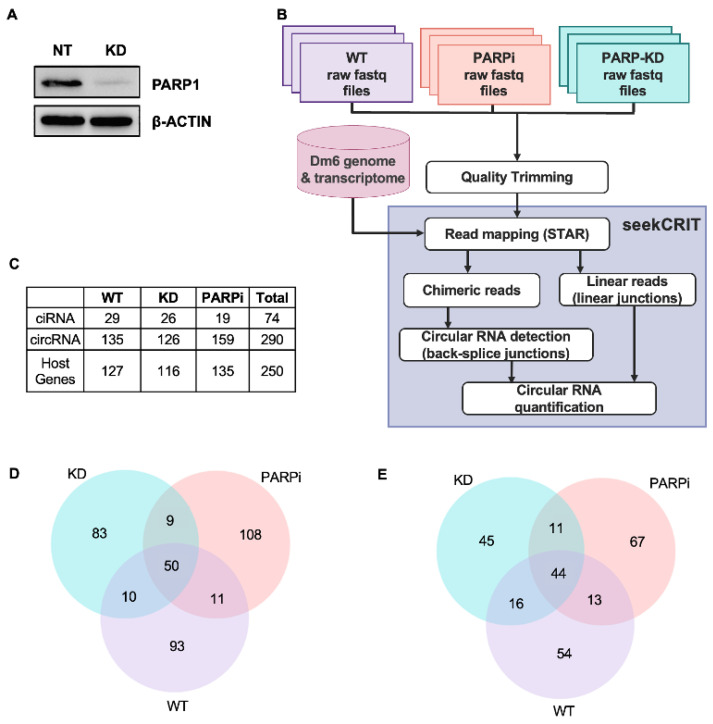
Detection of circRNAs in paired-end sequencing data from total RNA using seekCRIT. (**A**) Western blot analysis of PARP1 depletion by siRNA. β-actin used as loading control. (**B**) Schematic representation of seekCRIT computational pipeline [18]. (**C**) Number of intronic circRNAs and exonic circRNAs (“backsplice junctions”), discovered by seekCRIT and the host genes they come from in the various experimental conditions. (**D**) Overlap of circRNAs between WT, PARPi, and PARP KD conditions. (**E**) Overlap of the circRNA-producing genes (host genes) between WT, PARPi, and PARP KD conditions.

We then randomly chose several RNA-seq-detected circRNAs for validation using Sanger sequencing. The RNA was isolated and converted to cDNA, which was used for PCR amplification using specific primers targeting the unique backsplice junction of the circRNAs. As control, genomic DNA was also isolated and used for PCR (Appendix A). 

### 3.2. Possible Functional Implications and Identification of PARP1-Related circRNA–miRNA Interactions

An increasing number of studies show that circRNAs regulate their host gene expression at the transcriptional and splicing levels [27,29,30,31]. To begin to tease out the function of the PARP1-mediated and dysregulated circRNAs in PARPi and KD, we first analyzed whether PARP1-mediated circRNAs emanated from differentially expressed genes (DEGs), which would suggest regulation via differential expression. In total, we found that 299 genes were differentially expressed in PARPi vs. WT and 1435 in KD vs. WT (q < 0.05) (Figure 2A). Most of the dysregulated genes in PARPi conditions were not the same in KD conditions (Figure 2B). These results confirm our previous findings that the physical presence of PARP1 acts differently on gene expression compared to its PARylation activity [9]. Interestingly, of the 298 differentially expressed genes in PARPi vs. WT conditions, only 30 produced circRNAs in PARPi and 4 in WT conditions (Figure 2C). Similarly, even with the higher number of DEGs (1434) in PARP1 KD vs. WT (Figure 2A), only 20 and 22 of those produced circRNAs in KD and WT conditions, respectively (Figure 2D). These results suggest that most of the identified circRNAs produced in our experimental conditions do not seem to be regulating the expression of their host genes, as many of them were not differentially expressed. Gene Ontology analysis identified that the PARylation-regulated biological processes that are significantly altered are involved in developmental and morphogenesis processes (Figure 2E), while the translation, ribosome, and cellular respiration processes were identified as significantly altered in PARP1 KD (Figure 2F). Enrichment analysis on those DEGs also identified the dysregulation of spliceosomal components (Appendix A), which is known to affect backsplicing. These results are in line with our previous studies on PARPi and PARP1-regulated gene expression [9].

In addition to regulating the genes at the transcriptional initiation, PARP1 also regulates alternative splicing events [9,10]. The generation of circRNAs has been shown to compete with linear mRNA splicing [32]. As such, several studies have shown that skipped exons overlap with circRNAs [33,34]. We therefore asked whether our detected PARP1-mediated circRNAs resulted from skipped exon events. We used DEXSeq [22] and rMATS [23] to determine whether circRNA host genes are differentially spliced. We detected 471 and 422 genes that were alternatively spliced in PARPi and PARP1 KD conditions, respectively, with an overlap of 201 genes between PARPi and KD (Figure 3A).

We then asked whether backsplicing was detected in the alternatively spliced genes. We found that while there were 471 genes alternatively spliced due to PARPi, only 56 produced circular RNAs (Figure 3B). Similarly, while there were 422 genes alternatively spliced due to PARP1 KD, only 52 of the differentially spliced genes produced circular RNAs (Figure 3C). These results suggest that the circRNA host genes were not predominantly alternatively spliced. Nevertheless, we asked whether those circRNAs produced from alternatively spliced genes were generated from skipped exons. Therefore, we looked for instances where a circRNA was generated from a skipped exon event or for a circRNA that was lost due to a retained intron event. We did not find this to generally be the case. Out of the 422 KD differentially spliced genes, only 38 (9%) had a skipped exon event that also produced circRNAs in PARP KD (Appendix A). Similarly, 42 out of the 471 (8.9%) differentially spliced genes had a skipped exon that also produced circRNAs in PARPi conditions (Appendix A). Examples of Sashimi plots showing the overlay of skipped exons and backsplicing in WT, PARPi, and KD conditions are shown in Figure 3D. These results further support the finding that PARP1-mediated circRNAs are not ‘by-products’ of splicing but are true backsplicing events. 

Lastly, to further delineate the function of these circRNAs, we asked whether the generated circRNAs regulate gene expression by acting as miRNA sponges. Since very few miRNAs and their targets have been validated in the Drosophila genome and because of the high phylogenetic conservation of microRNA sequences [35], we used the human miRNA sequences as seeds to determine potential miRNA target sites present within our discovered circRNAs. Several microRNAs were detected as possible targets for the unique circRNAs produced in WT, PARPi, and KD conditions (Appendix A). We used the top possible 100 microRNA targets for GO biological processes (Figure 4) and KEGG pathway analysis (Appendix A) to determine the functional impact on circRNA–microRNA targeting. These analyses showed that circRNAs generated in WT conditions could target microRNAs important in development, differentiation, and cell cycle processes. Furthermore, the possible targets for the circRNAs generated in PARPi conditions are involved in very similar processes as those for WT-generated circRNAs, such as development, adhesion, and cell cycle. On the other hand, the KD-generated circRNAs are involved in other pathways such as RNA biogenesis, translation, and chromatin remodeling (Figure 4). In summary, the possible microRNA targets of circRNAs generated in WT and PARPi conditions seem to favor processes and pathways involved in development and differentiation, supporting PARP1’s role in proper development (Figure 4 and Appendix A), while the potential KD circRNA–microRNA pathways target many RNA processes including ribosome and RNA processing and splicing (Figure 4 and Appendix A), supporting PARP1’s role in splicing regulation, a key component of circRNA biogenesis. 

### 3.3. Characterization of Host Gene Architecture

Gene architecture and sequence bias, together with epigenetic marks, dictates the space–time (Figure 5A) of co-transcriptional splicing [36] and backsplicing [37,38,39]. To explore whether PARP1 plays a role in this space–time model, we profiled the gene architecture (gene, exon, and intron lengths) and the sequence bias in host genes and circRNAs found in PARP1 WT and compared these to those generated in PARPi and KD conditions. The findings are summarized in Figure 5. 

Gene Length: The host genes of circRNA-producing genes are, on average, longer and contain more splice sites than other genes [14]. We therefore compared the length of the genes in WT, PARPi, and KD conditions. We found that the genes producing circRNAs in all the conditions were, on average, longer than other genes (Figure 5B). However, there were no significant differences in the gene length between the conditions. These results support earlier findings that circRNA-producing genes are generally longer [14,40,41]. 

Intron Length: circRNA-producing host genes have atypical exon and intron lengths, with the exons within the circRNAs generally being shorter and the introns adjacent to the backsplice sites being exceptionally long [41,42]. We asked whether our PARP1-mediated circRNAs follow this principle. First, we analyzed the introns and observed that in PARP1 WT conditions, the flanking upstream and downstream introns of the circRNAs are, on average, 9131 bp and 9355 bp long, respectively (Figure 5C; Appendix A). These introns are longer than the typical Drosophila intron, which has an average length of 1832 bp (Figure 5C). The circRNAs in PARPi conditions were flanked upstream and downstream by very long introns of average lengths of 14,395 bp and 8228 bp, respectively (Figure 5C). In KD-producing circRNAs, the upstream introns were longer on average (11,251 bp) than WT upstream introns (9131 bp). Conversely, the downstream introns were shorter on average (6089 bp) than the WT downstream introns (9355 bp) (Figure 5D). Interestingly, while the WT-produced circRNAs had symmetrical flanking intron lengths, the depletion of PARylation and PARP1, resulted in circRNAs with introns of lopsided lengths, with the upstream intron almost twice as long as the downstream intron (Figure 5D). We made another interesting observation when we compared the lengths of the internal introns for the multiple exonic circRNAs. While the internal introns of the circRNAs produced in WT and PARPi circRNAs were of average lengths of 3683 bp and 4431 bp, respectively, the internal introns of the circRNAs produced in KD conditions were shorter and were consistent with the average length of the introns in Drosophila (Figure 5C). In general, our results are in line with previous studies showing that flanking introns of circRNAs are generally longer [14,41,42].

Exon Length: In general, the exons participating in circularization in all conditions are shorter than the average Drosophila exon, with an average length of 292 bp. In detail, WT exons were 372 bp long on average, and PARPi and KD were 292 bp and 351 bp long on average, respectively (Figure 5E). Since single exonic circRNAs tend to have longer exons than multiple exonic circRNAs, we further analyzed the average exon length of single exonic circRNAs compared to the average exon length of multi-exonic circRNAs. This analysis supported previous studies [41] showing that the exons of multi-exonic circRNAs are shorter on average than those from single-exonic circRNAs (Appendix A).

Based on the gene architecture, it seems that WT-generated circRNAs have similar gene features as PARPi, apart from the lopsided intron length (longer upstream introns and shorter downstream introns). The KD of PARP1 produced circRNAs from genes with very different architectural features generally compared to circRNAs produced in WT and PARPi conditions. 

Sequence Composition: Sequence bias includes G-quadruplex content, inverted repeats, and RNA protein-binding motifs. To begin to understand whether the PARP1 regulation of circRNA biogenesis occurs via cis base pairing of sequences and/or in trans by recruiting RNA-binding proteins (RBPs), we used a comprehensive analysis in determining the sequence bias in the introns flanking the circRNAs. We first asked whether repeat elements (REs), which have also been implicated in circRNA biogenesis, could be important for PARP1-mediated backsplicing and circularization. Since *Alu*-transposable elements, implicated in circRNA biogenesis, are absent in the Drosophila genome [43], we focused our analyses on other REs, including Drosophila-transposable elements [44]. We observed that, in general, there is an enrichment of satellite repeats, LINEs, and LTRs, and a loss of simple repeats and low complexity sequences in the flanking introns of circRNAs in all conditions compared to all genomic intronic sequences. However, some differences in the enrichments were detected. In WT conditions, there was a slight enrichment of satellite repeats in downstream introns compared to the upstream introns, while LINEs were more enriched in upstream introns compared to downstream introns (Figure 6A). PARPi showed enrichment of satellite repeats in upstream compared to downstream introns, while LINEs showed more enrichment in downstream introns compared to upstream introns (Figure 6B). In KD conditions, the results were more dramatic. The flanking introns of the circRNAs generated are not only devoid of rolling circle repeats but are highly enriched with satellite repeats compared to other Drosophila introns (Figure 6A–C). In addition to satellite enrichment, these introns were more enriched in LINEs with a bias of upstream enrichment compared to the downstream introns. In summary, compared to WT and PARPi, PARP KD flanking introns were most enriched for satellite repeats devoid of LINEs. These results suggest that the circRNAs generated in the presence of PARP1 use different repeats compared to non-PARP1-regulated circRNAs.

Like REs, which help form pre-mRNA looping through base pairing in flanking introns, RNA binding proteins (RBPs) also mediate the biogenesis of circRNAs by binding to flanking introns in trans and homodimerization via protein–protein interactions [45]. Since PARP1 recruits splicing factors [9] and DNA repair proteins [46,47], we asked whether PARP1 might be mediating circRNA biogenesis via the recruitment of RBPs. Due to the lack of well-documented RNABPs in Drosophila, we analyzed sequences surrounding our identified circRNAs for enriched motifs of known human RBPs compared to background. Specifically, we analyzed those RBPs whose motifs were found symmetrically within the introns flanking the 5′ and 3′ backsplice sites. The motifs for several RBPs were found significantly (*p* < 0.05) enriched in each of the conditions (Figure 6D). Some of these RBP motifs were also unique for each condition, while some showed overlap in the conditions. Interestingly, the motifs for QKI, ANKDH1, and SRSF10 overlapped in the WT and KD conditions (Figure 6D). Since PARylation impacts the function of SFs, we also analyzed the binding sites in PARPi host genes. The motifs for seven RBPs were enriched in these introns (Figure 6D) and only three of these motifs were unique to the PARPi condition. The others were shared in the other conditions—three RBPs were shared with WT host genes and only one RBP motif (RBM28) was shared between the PARP KD and PARPi conditions. These results again suggest that the effects of the presence of PARP1 (WT and PARPi) are more common than in KD conditions. The presence of QKI and ANKHD1 motifs in all three conditions indicates their importance in general circRNA biogenesis [48,49].

Since PARP1 is known to bind G-quadruplexes [50], and G-quadruplexes are enriched at splice sites to modulate alternative splicing [51], we considered the possibility that G-quadruplexes may be differentially enriched within PARP1-regulated host genes. Surprisingly, we found a reduction of G4 within all host genes’ introns identified in our model compared to background (Figure 6E). However, studies have reported reduced GC content as a feature of host genes, so we wanted to ensure that our observation of reduced G-quadruplexes was not an artifact of nucleotide composition. We first looked for differences of AT/GC content (Appendix A) within our host genes compared to background and found no significant differences (Appendix A). We then looked for individual nucleotide bias in these genes and, interestingly, found the enrichment of A and C nucleotides, with a concurrent depletion of Ts (Appendix A). This trend was exaggerated in the PARP KD samples, although was not found to be significantly different from the WT host genes. Nevertheless, there was no difference in the density of G nucleotides (Appendix A), despite the depletion of G-quadruplex motifs. We also profiled the intron-exon boundaries of the circRNA splice sites. We found no significant differences in GC content within the 500 bp intronic regions upstream and downstream of the backsplice site in the WT conditions compared to the PARPi and KD conditions (Appendix A).

### 3.4. Profiling of Host Gene RNAPII Pausing

Gene architecture regulates the movement of RNAPII in transcription to mediate co-transcriptional splicing. We previously showed that PARP1, as an alternative splicing mediator, regulates RNAPII pausing within gene bodies [8], specifically at exon–intron boundaries. We wondered whether the PARP1-mediated pausing of RNAPII could also regulate backsplicing. For this, we utilized our previously acquired NET-seq dataset, GSE118266 [10], to determine whether PARP1 regulates pausing at different regions within the host genes. We observed remarkably distinct RNAPII pausing profiles in WT host genes compared to KD host genes (Figure 7). In the WT host genes, which have more conserved-like circRNA gene architecture (Figure 5A), there is generally a high accumulation of RNAPII along these genes compared to KD genes. In detail, consistent with our studies [10] and others [41], there is significant pausing of RNAPII at the promoters of these genes, and at exon–intron boundaries [41,52]. Of interest to circRNA biogenesis, we also observe RNAPII pausing earlier in the gene, at exon 1 and at the acceptor exons of the circRNAs, relative regions downstream of the circRNA (Figure 7A,B). In contrast, the pausing in KD host genes was significantly less compared to WT host genes (Figure 7A–D). Additionally, in the host genes in KD conditions, RNAPII accumulated in the downstream region of the circRNAs (Figure 7A,D). When PARP1 was knocked down, the pausing of RNAPII in the WT host genes was decreased, except for regions found internal to the circRNA (Figure 7E). On the other hand, pausing was generally increased in the KD host genes when PARP1 was depleted (Figure 7F). These results suggest that the depletion of PARP1 in KD host genes may increase the speed of RNAPII along the gene body and increase the resident time of RNAPII downstream of the backsplice donor, to favor backsplicing in these genes with less favorable gene architecture. 

### 3.5. PARP1 Regulates the Transcriptional Output of circRNA Host Genes (NanoString)

Since PARP1 regulates both transcription initiation and splicing, we next asked whether PARP1 regulates the types of isoforms from a particular gene. To answer this question, we calculated the circular to linear ratio of the circRNA host genes. For this analysis, we chose circRNAs that were common between the treatment groups and used the direct digital counting of RNA molecules by NanoString technology. This method bypasses the problems of concatemerization during reverse transcription in qRT-PCR (Figure 8A). We designed a custom codeset targeting the unique backsplice junctions for circRNA detection and another codeset targeting the mRNA regions not included in the circular products to detect host gene linear mRNAs (Figure 8A). This setup allowed us to simultaneously quantify both the linear and circRNA transcripts from 26 host genes in WT, KD, and PARPi conditions. First, we calculated the expression of these genes in WT and compared them to PARPi and KD conditions. Normalized raw counts were used to measure the fold change between PARPi and KD to WT conditions. We showed that the inhibition of PARylation activity resulted in the fewest changes in host gene expression, with only seven host genes (27%) altered (Figure 8B). The knockdown of PARP1 had a greater impact, altering the expression of 18 host genes (69%) (Figure 8C). We reasoned that since knockdown is not knockout, the residual PARPi might still have PARylation activity. We therefore performed similar experiments whereby the knockdown cells were treated with PARP1 inhibitors (KD + PARPi). This combination of PARP1 KD and PARPi, as expected, resulted in the greatest degree of change in host gene expression (73%) (Figure 8D). Three host genes (11%) remained unaltered regardless of treatment. These results suggest that the presence of PARP1 is more important in PARP1-mediated gene expression from these genes than PARylation inhibition. Second, we investigated whether PARP1 mediates the overall gene transcriptional output. We analyzed the ratio of changed circRNA expression to the change in linear mRNA expression from each host gene in the PARPi, PARP KD, and combined treatments (Figure 8E–G). We plotted the log2fold change of each circRNA expression compared to the log2fold change of its host linear mRNA. While a change in the expression of most of the circRNAs correlated positively with linear mRNA expression, most did not follow a 1:1 ratio change. These results indicate that there is a splicing regulation and not generally a transcription initiation regulation. We found that 19% (five) of the genes showed an increased C:L ratio, 50% (thirteen) showed a decreased C:L ratio, and 31% (eight) showed no significant change in expression due to PARP1 KD (Figure 8E–G). These results suggest that PARP1 could be regulating transcriptional output and changing the types of transcriptional events generated. 

A recent study showed that the differential movement of RNAPII between introns and exons dictates a C:L ratio and ultimately the transcriptional output [41]. To understand whether PARP1 could be regulating differential exon/intron RNAPII movement and changing the transcriptional output, we measured the PARP1-mediated pausing within the introns and exons of the host genes. Since most circRNA measured consisted of single exons, very little pausing was detected overall for introns or exons within the backsplice sites (65.5% no change for exons, 79.3% no change for introns) (Appendix A). In general, we observed increased pausing within the host gene exons (57.7% increased pausing compared to 11.5% decreased pausing and 30.8% no change). The pausing changes within introns upon PARP1 KD were more bimodal/diverging with increased (53.8%) or decreased pausing (34.6%). Only 11.5% of the host gene introns showed no change. Nearly all the genes with altered exon pausing also had changes in intronic pausing (94.5%) (Figure 8H). We next profiled the pausing within the host gene introns and exons for those genes with an increased C:L ratio and compared them to those with a decreased C:L ratio. Interestingly, in both cases, the exons showed increased pausing; however, in the genes that had an increase in the C:L ratio, the introns showed decreased pausing, whereas the genes with a decrease in the C:L ratio showed increased pausing within the introns as well (Figure 8I). These results suggest that PARP1 regulates differential RNAPII movement between exons and introns to elicit specific transcriptional output. 

## 4. Discussion

Our previous studies [9,10] and the studies of others [53,54] showed that PARP1 regulates splicing. Specifically, we showed that PARP1 directly regulates co-transcriptional splicing, whereas PARylation acts indirectly by PARylating and activating splicing factors. In the present study, we now show that PARP1’s role in splicing regulation extends to backsplicing, as we detected many unique circRNAs in WT, PARPi, and PARP1 KD conditions. We also hypothesize that PARP1 regulates circRNA biogenesis through the same co-transcriptional mechanisms by which it regulates linear splicing. 

The low expression levels, the shared sequence identity of circRNAs with their cognate linear mRNAs, and the fact that they can only be detected via the backsplice junction reads makes detection and quantification challenging. Though several bioinformatic detection tools have been developed as well as the enrichment of circRNAs after RNAse digestion, the detection and quantification of circRNAs still suffers from low accuracy and high false-positive rates [55,56]. In our study, we used seekCRIT [18], a computational tool that identifies all transcripts, including circRNAs, from high, in-depth sequencing of ribosomal-depleted RNA. Even with these high sequencing reads, we were not able to detect many significant changes in the expression of circRNAs caused by PARylation inhibition or PARP1 depletion. Thus, to understand the impact of PARP1 and its PARylation activity on circRNA biogenesis, we focused on the circRNAs that were uniquely detected and analyzed the regulatory elements that might aid in their biogenesis and how PARP1’s presence might facilitate their biogenesis. 

In general there is no consensus mechanism for the biogenesis of circRNAs [57]. It is widely understood that cis and trans elements mediate circRNA biogenesis, bringing circRNA-forming exons into proximity [58,59]. For instance, cis-regulatory elements such as intronic complementary or repeat sequences and non-complimentary sequences or RNA binding proteins sites [60] may create secondary structures that facilitate back splicing. To test, we profiled cis elements such as the length of flanking introns, repeat elements, GC content, and RNA-binding motifs surrounding the unique circRNAs generated in WT, PARPi, and KD conditions. We found interesting patterns, as follows. (1) The flanking intron length for WT circRNAs was symmetrical, while in PARPi and KD, the upstream introns were relatively longer compared to the downstream intron length. We therefore hypothesize that circRNAs generated after the depletion of PARP1 or its enzymatic activity require longer 5′ flanking intronic sequences, providing enough resident time for RNAPII movement, a critical regulator for alternative splicing [61,62] and backsplicing [41,63]. The differences in the intron lengths between the conditions may also introduce more complementary repeats or RBP motifs, which could aid in promoting exon circularization [59]. (2) We profiled repeat elements and found that while the flanking introns in all conditions were enriched in transposable elements (LINES and LTRs) and satellite repeats, they were depleted in simple repeats (Figure 6A–C). However, some differences in the enrichment level between the conditions were observed. Of particular interest was the differential presence of helitrons such as Drosophila interspersed elements, or DINES, and miniature inverted-repeat transposable elements, or MITEs, between the treatment groups (Figure 6A–C). They were more enriched in PARPi conditions than in wild type host genes and were reduced in PARP KD host genes (Figure 6A–C). While we do not yet understand the significance of this difference in enrichment, it is possible that the enrichment of satellite repeats in our host genes is due to intron expansion during evolution, which is a known function of satellite repeats [64]. However, further research will be needed to explore this possibility and the significance of this association. (3) We also analyzed the GC content and G-quadruplexes, which fold into complex structures, and both are important for splicing regulation [65] and are also bound by PARP1 [53,66]. While we observed differences in the GC content between introns and exons, there was no significant difference between the conditions (Appendix A). G-quadruplexes are enriched at splice sites to modulate alternative splicing [51] and can be bound by PARP1 [67], possibly with consequences in circRNA biogenesis. While a general depletion of G-quadruplexes was found within the flanking introns of circRNAs in all conditions, this depletion was exaggerated in PARP1 knockdown (Figure 6E). No studies have yet shown the importance of G-quadruplexes in the formation of circRNAs. This finding is very likely not an artifact of low GC content in the host genes of circRNAs since we found no difference in the G-density in these regions (Appendix A). We hypothesize that, most likely, the reduction in G-quadruplexes near circRNAs might be evolutionary and requires further studies to understand the functional relevance. 

Is PARP1 acting to mediate the time needed for RNAPII to move along the gene body? Just as shown previously, the flanking introns of circRNAs in all conditions were long [41]. In WT, we observed that the flanking introns were symmetrically longer, whereas in the depletion of PARP1 or its enzymatic activity, the upstream introns were longer compared to the downstream introns. Long introns introduce the time needed for RNAPII movement, which is critical for splicing, presenting a “window of opportunity” for RNAPII and its associated splicing factors to decide the splicing decisions [68]. This would be especially true for backsplicing because the upstream acceptor must still be available by the time the downstream donor is synthesized. We tested how the PARP1–chromatin structure mediates this window of opportunity for RNAPII to regulate circRNA biogenesis. Our analysis shows PARP1-mediated RNAPII proximal pausing as well as pausing at the upstream acceptor exon of the circRNA (Figure 7A). This is in line with our studies and those of others showing PARP1’s role in RNAPII pausing at the promoter [7,9] and at intron/exon boundaries [10]. A comparison of this RNAPII pausing within these same genes in KD conditions, where no circRNAs are produced, showed the loss of this proximal and intron/exon boundary pausing. However, a strong pausing is seen within the exons that would have otherwise formed circRNAs in WT conditions. We speculate that this change of RNAPII movement in the KD condition then promotes linear splicing. This is an interesting idea, which is also favored when we analyze the pausing patterns in KD conditions. In KD conditions, where circRNAs are produced in the absence of PARP1, we observed low pausing in the proximal promoter and circRNA acceptor regions, and a marked accumulation of RNAPII pausing downstream of the circRNA at the immediate downstream intron/exon of the circRNA donor (the end of the circRNA transcript). These results are consistent with PARP1’s role in RNAPII elongation—a lack of PARP1 causes the movement of RNAPII towards the 3′ end of the gene. 

How are circRNAs made in the absence of PARP1? We hypothesize that the downstream pausing observed in the KD conditions compensates for the reduced 3′ flanking intron length. Just like with WT, where the longer introns create a window of opportunity for the time needed for RNAPII to move further into the gene body, and the increased pausing downstream in PARP1 KD equally affords such time needed for circRNAs to be generated. These studies show that PARP1 mediates a chromatin structure that aids in creating the window of opportunity for RNAPII and its associated factors to make the splicing decisions. If this is the case, we analyzed the impact of PARP1-mediated RNAPII movement in host exons and introns on transcriptional output (circular to linear ratio). We used NanoString to measure the circRNA expression alongside the linear expression. We observed that there is a general increase in the pausing in exons. However, when the increase in exon pausing was accompanied by an increase in intron pausing, more circRNAs were made from the gene. On the other hand, when exon pausing was accompanied by decreased intron pausing, more linear transcripts were made from the same gene. Our results are consistent with those of the circular to linear ratio [63]. Overall, our results support the role of PARP1 in regulating the dynamic RNAPII pausing in backsplicing [41], but show that PARP1 fine tunes the movement of RNAPII between host gene exons and introns for transcriptional output decisions. 

Taken together, these results show that PARP1 regulates not only regular splicing, but also backsplicing. We also show that the interplay of certain cis features and trans features and PARP1 regulate backsplicing. We further show that by binding to the chromatin structure, PARP1 modulates not only the rate of RNAPII, but also the dynamic movement within different regions of the gene for specific transcriptional output decisions. More studies will be needed to test the direct effect of PARP1 on RNAPII movement in circRNA host gene bodies. These studies, however, provide a first glimpse of PARP1–chromatin function in modulating backsplicing decisions, providing a platform to directly mechanistically examine the factors that mediate this function of PARP1.

## Figures and Tables

**Figure 2 cells-12-01160-f002:**
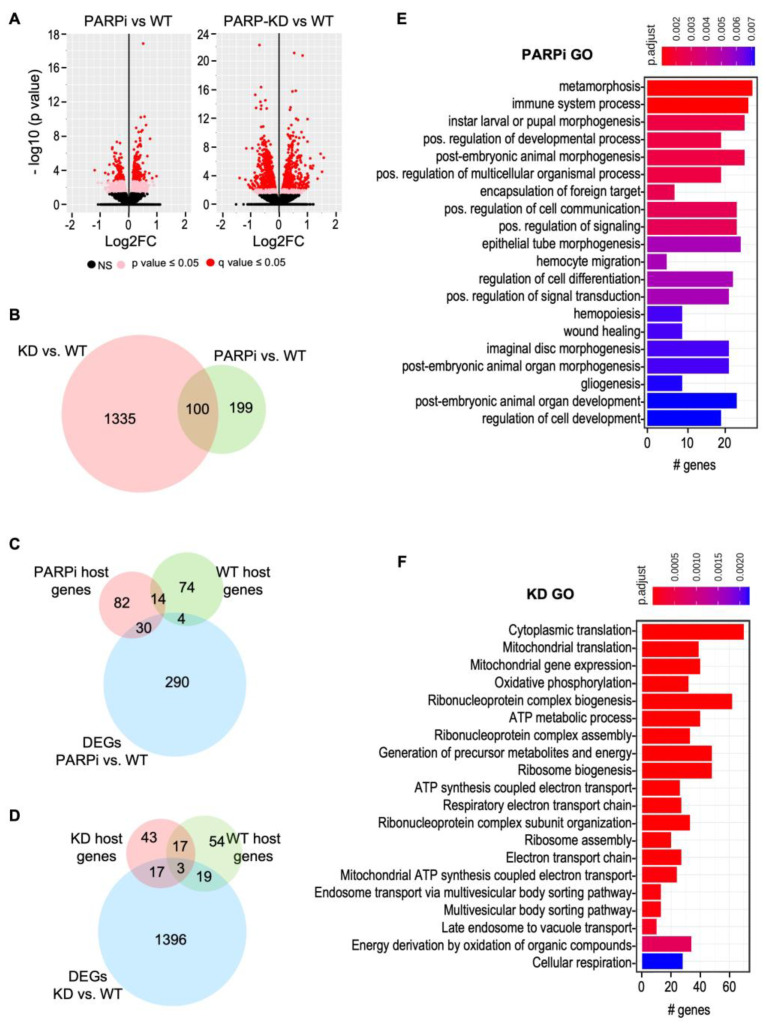
Gene regulation mediated by PARP1 and PARylation. (**A**) Volcano plots showing differentially expressed genes in PARPi and PARP KD vs. WT conditions. Y-axis is -log *p*-value and x-axis is the log2 value of the fold change. Each dot represents one gene. Black dot represents no significant difference, pink represents *p* < 0.05, red dot represents q value < 0.05. (**B**) Venn diagram showing the DEGs that overlap between PARP KD (pink) compared to inhibition of its PARylation activity (green). (**C**) Venn diagram showing PARPi host genes (red) compared with WT host genes (green) and genes differentially expressed in PARPi (blue). (**D**) Venn diagram of PARP KD host genes (red), WT host genes (green), and genes differentially expressed in PARPi (blue). Numbers depicted in the intersections between circles represent the numbers of genes that are commonly regulated in two, three, or four conditions. (**E**,**F**) Top 20 enriched GO biological processes ranked by significance for PARP1 (**E**) and PARP KD (**F**).

**Figure 3 cells-12-01160-f003:**
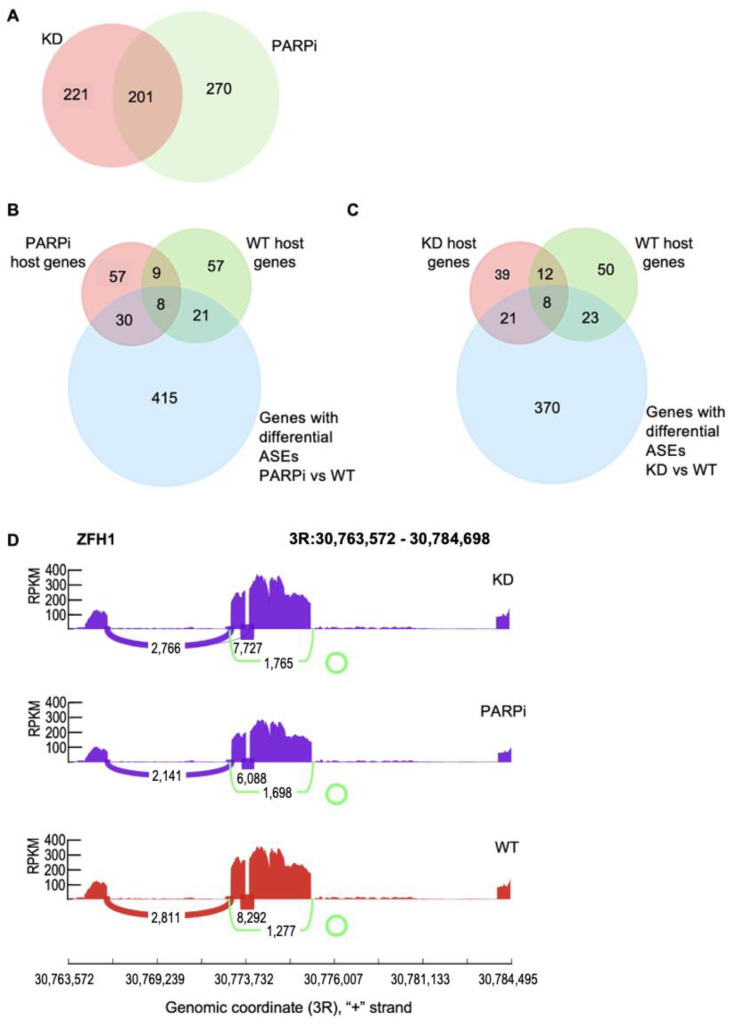
Global alternative splicing events mediated by PARP1 and PARylation. (**A**) Venn diagram showing the differentially spliced genes detected by MATS in PARP1 knockdown compared to PARylation inhibition (PARPi). (**B**) Venn diagram of the host genes of the circRNAs generated in WT and PARPi over the differentially spliced events in PARPi compared to WT conditions (*p* < 0.01, by Student’s *t*-test). (**C**) Venn diagram of the host genes of the circRNAs generated in WT and PARP KD over the differentially spliced events in PARP1 KD compared to WT conditions. PARP1-mediated ASEs (alternative splicing events) detected by rMATs (*p* < 0.01, by Student’s *t*-test). (**D**) Sashimi plots of one example of a host gene, ZFH1, showing alternative splicing events and circRNAs formed in each condition.

**Figure 4 cells-12-01160-f004:**
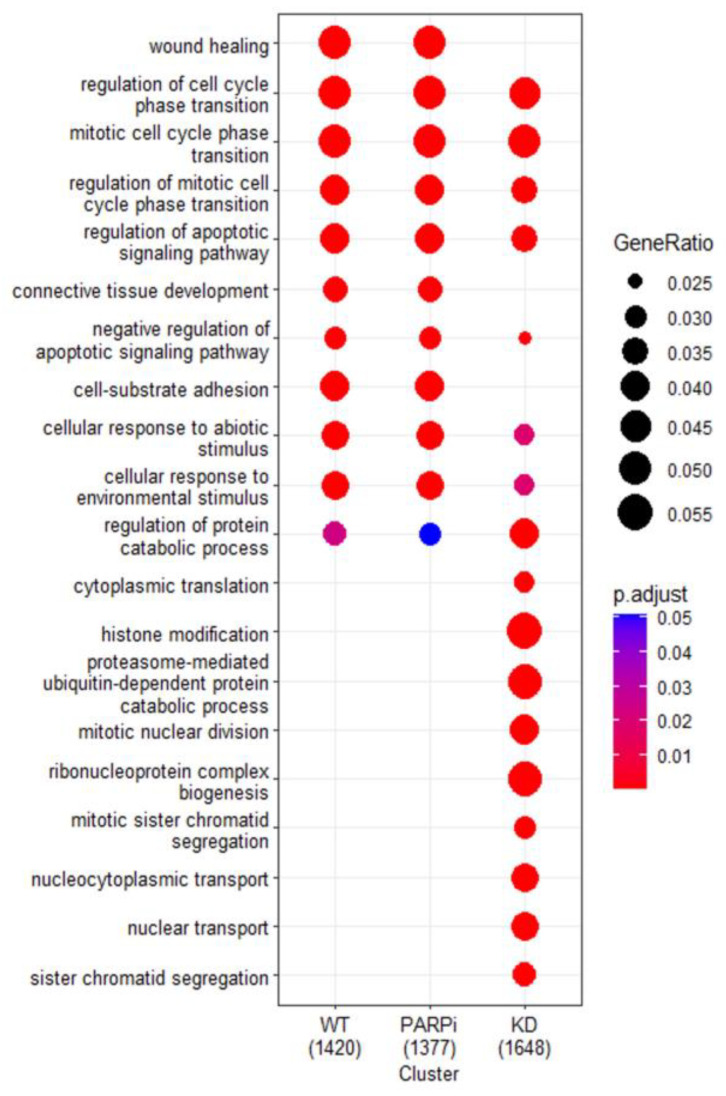
Prediction of PARP1 and PARylation circRNA–miRNA–mRNA regulatory network. Dot plot comparison of differential GO biological processes regulated by predicted circRNA–microRNA interactions for circRNAs unique for WT (column 1), PARPi (column 2), and PARP1 KD (column 3). The most frequent miRNA targets in each experimental circRNA group were identified from miRTarBase (Release 9.0) [24] relative to human miRNA seeds. The target functional annotation was completed using clusterProfiler.

**Figure 5 cells-12-01160-f005:**
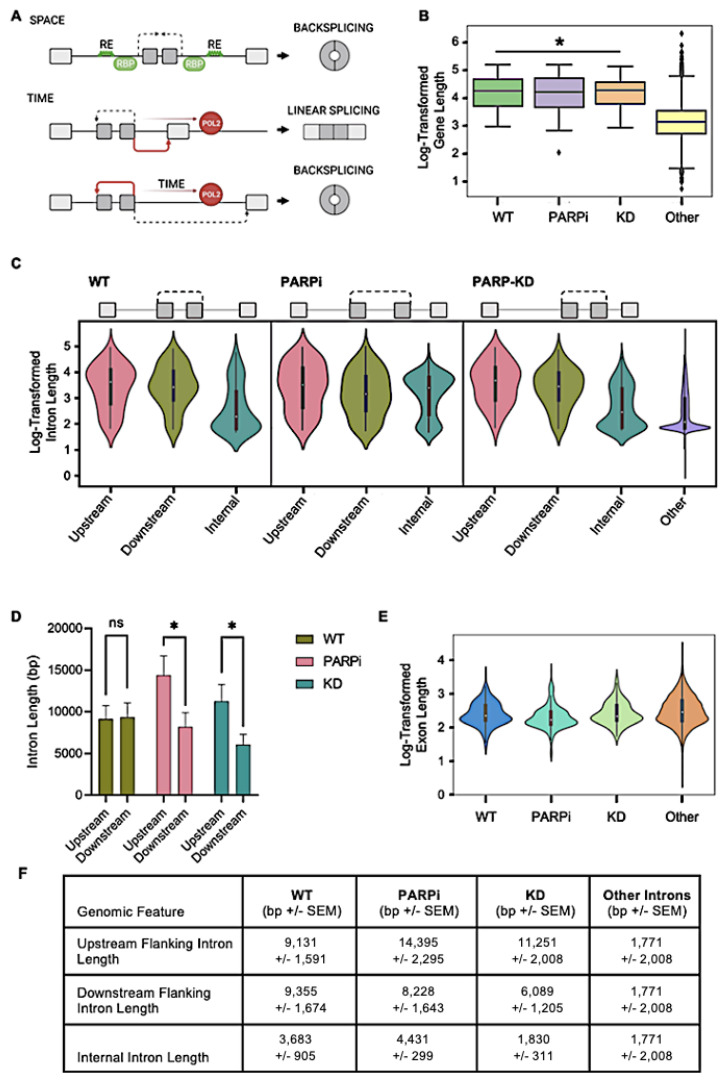
Linear architecture of circRNA host genes is altered upon PARP1 disruption. (**A**) Schematic diagram describing the influence of transcriptional space and time on backsplicing. Boxes represent exons, lines represent introns, factors regulating transcriptional space are green, RNAPII is red. Repeat elements and RBPs in flanking introns bring backsplice sites into close physical proximity to promote backsplicing. The prolonged time it takes for RNAPII to travel to a downstream acceptor provides a window of opportunity for backsplicing to occur. (**B**) Mean length of WT (green), PARPi (purple), PARP KD (orange) host gene lengths compared to other Drosophila genes (yellow) (* *p* < 0.01 vs. Other). (**C**) Lengths of upstream introns (pink), downstream introns (green), and internal introns (teal) compared to other Drosophila introns (purple) in WT (left), PARPi (middle), and PARP KD (right). (**D**) Bar graph highlighting the difference in upstream vs. downstream intron lengths (bp) in WT (green), PARPi (pink), and PARP KD (teal). Students *t*-test comparing upstream to downstream (* *p* < 0.05, error bars represent SEM). (**E**) Lengths of exons in WT (blue), PARPi (teal), and PARP KD (green) compared to other Drosophila exons (orange). (**F**) Table summarizing the mean lengths shown in the figures.

**Figure 6 cells-12-01160-f006:**
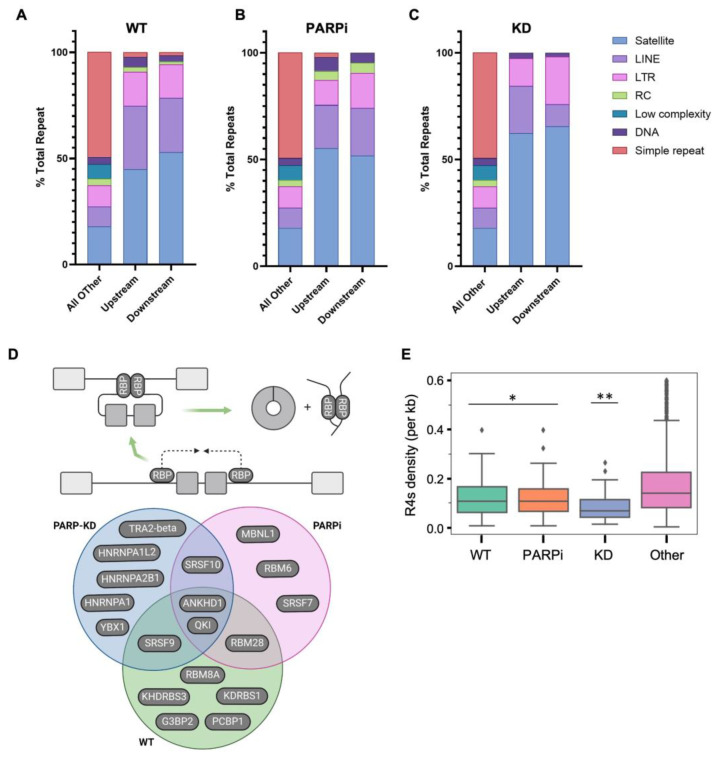
Flanking introns are differentially enriched for elements that regulate transcriptional space. The flanking introns of circRNA host genes were differentially enriched with repeat elements. Repeat elements enriched in wild type (**A**), PARPi (**B**), and PARP KD (**C**) upstream and downstream flanking introns compared to other Drosophila introns by type. (**D**) Venn diagram comparing RBP motifs significantly enriched (*p* < 0.05) within both flanking introns of WT, PARPi, and PARP KD. The presence of unique sets of RBP motifs in flanking introns suggests PARP1 circRNAs are regulated by different pathways. (**E**) Box and whisker plot showing the density of R4 G-quadruplexes in WT, PARPi, and KD host genes compared to other Drosophila genes. The results (bar graph) are represented as mean plus SEM (*n* ≥ 3; Student’s *t*-test, * *p* < 0.05, ** *p* < 0.05). The G-quadruplexes in the host genes and other Drosophila genes were identified using quadparser [26].

**Figure 7 cells-12-01160-f007:**
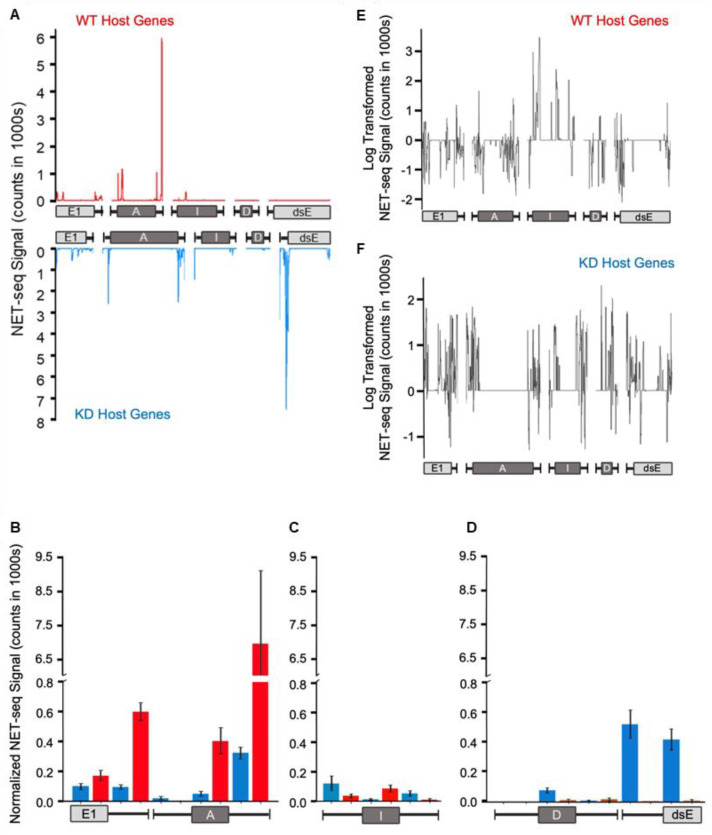
circRNA host genes in PARP KD exhibit distinct pausing profiles compared to WT circRNA host genes. NET-seq data were used to measure RNAPII pausing in WT and PARP KD conditions. Pausing was measured at exons (boxes) and 300 bp upstream and downstream of exons (solid lines). For each set of host genes, pausing within gene bodies was measured at exon 1 (E1) upstream of circRNA exons, the circRNA acceptor exon (**A**), the circRNA internal exons (I), the circRNA donor exon (**D**), and the exon downstream of the circRNA region (dsE). (**A**) RNAPII accumulates early in WT host genes (top) compared to pausing within PARP KD host genes (bottom) where RNAPII stalls at the dsE. (**B**–**D**) Bar chart showing a side-by-side comparison of the differences in RNAPII accumulation at specific host gene regions as described in (**A**). The differences in RNAPII pausing upstream of the acceptor and acceptor regions of the circRNA are shown in (**B**), while (**C**) and (**D**) show differences in RNAPII pausing internal to the circRNA and downstream of the donor exon of the circRNA, respectively. (**E**,**F**) The change in RNAPII pausing within WT (**E**) and PARP KD (**F**) host genes upon PARP1 depletion.

**Figure 8 cells-12-01160-f008:**
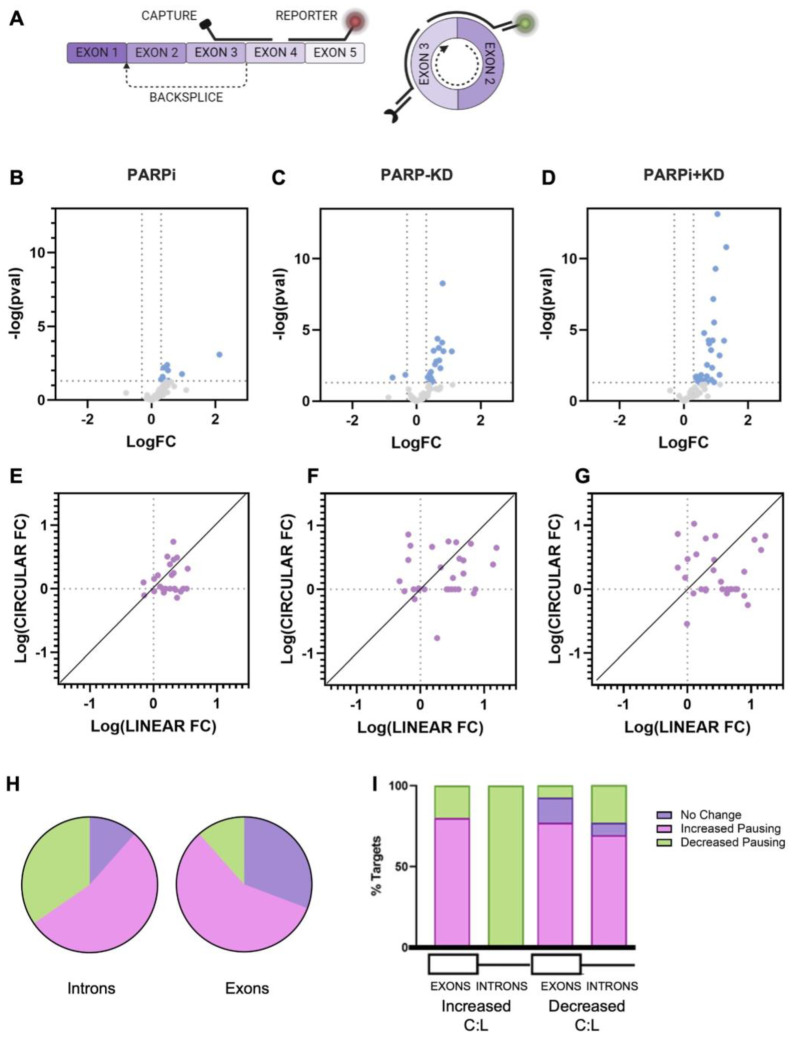
PARP1 fine tunes host gene transcriptional output through RNAPII pausing within host gene introns. (**A**) Schematic diagram of the NanoString codeset design. RNA transcripts are detected with complementary capture and reporter probes designed to hybridize the linear splice and circular backsplice junctions, respectively, for each host gene. Transcripts from ribosomal-depleted paired-end RNA-seq were counted via a unique fluorescent barcode assigned to each reporter probe. (**B**–**D**) Volcano plots showing the logFC of host gene expression due to PARPi (**B**), KD (**C**), and PARPi + KD combination treatment (**D**). Each dot represents either an mRNA or a circRNA transcript. Gray dots logFC < 2, *p* > 0.05; blue dots logFC > 2, *p* < 0.05. The log fold change of the circular transcripts was plotted against the log fold change of the linear mRNA for each host gene in PARPi (**E**), KD (**F**), and PARPi + KD combination treatment (**G**). (**H**) Pie charts showing the proportion of host genes with increased pausing (pink), decreased pausing (green), or no change in pausing (purple) within introns (left) or exons (right) upon KD. (**I**) Stacked bar graph showing the change in pausing within exons or introns of host genes. Host genes with increased circular to linear (C:L) ratio were compared to host genes with decreased C:L ratio.

## Data Availability

The analyzed data have been deposited in GEO with accession number GSE225445.

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
