# Peer review of "PARP1 Regulates Circular RNA Biogenesis though Control of Transcriptional Dynamics"

_cells, 2023, doi:10.3390/cells12081160_

Round 1
Reviewer 1 Report
About the paper:
circRNAs are a comparatively new class of RNAs to be discovered and have been shown to play an important role in cellular pathways, thus affecting health and disease. Our understanding of most of the processes underlying circRNA production and function is inadequate, though ever increasing. The paper ‘PARP1 regulates circular RNA …. dynamics’ is an attempt in that direction and uses RNA-Seq and bioinformatics approaches to hypothesize how the production of circular RNAs is affected in/due to the presence or absence of PARP1 protein. The authors have conducted extensive analyses to drive home their point. RNA-Seq data were generated from S2 cells having wildtype PARP1, knocked down PARP1, and those treated with PARP inhibitor. This was followed by circRNA detection and differential expression analysis of genes, annotation, etc.
General comments:
1. The introduction is well-written and on point.
2. The Results section is way too detailed and can be modified to move details about protocols or findings to the Methods or Discussions section, respectively. For example, see lines: 171-175, 204-213, 252-256, 303-309, etc. This, I think, could increase the readability of the manuscript.
3. Many of the findings in the Results section can be summarised in tabular or picture format instead and the implications of these findings could be discussed in the Discussion section. For example, see subsection 3.3.
Minor issues:
1. Line 84: Is it ‘heat-activated’ or ‘heat-inactivated’?
2. Line 90: remove ‘of’.
3. Line 493: remove ‘NET-seq peaks’
4. Lines 212-215: It should be Fig. S2 instead of Fig. 2.
5. Lines 247-248: Sentence not clear.
6. The western blot given in Fig. S1 should be published as part of the main manuscript, not in the supplementary files.
7. Original pictures of gels in the supplementary files have not been uploaded.
Major issues:
1. The western blot in the supplementary file doesn’t match the original picture of the image provided. Please check.
2. The numbers in the Venn diagrams (Fig. 1C, D) don’t add up. Please check. Fig. 1B, what does row 1 represent?
3. The marker lane in the WB seems to be pasted from another blot.
4. The most important issue with the paper is that the authors have sequenced the total RNA, instead of going for the enrichment of the important fraction (i.e. circRNA).
5. Since circRNAs occur in a very low proportion, I don’t think the results are reliable enough in the absence of RNAse R digestion or RAPD. This is complicated by the fact that the authors have centered all of their analysis around the circRNA fraction while sequencing total RNA and skipping almost the entire part about mRNAs.
6. Also, since I was not able to access the raw FASTQ files and the authors have not mentioned the sequence depth/coverage, I cannot appreciate how truly the findings represent the actual cellular/molecular picture. A table showing how much of the sequence data aligned with each RNA fraction (mRNA, circRNA, other non-coding RNA types) would have been of immense help. Whether the depth of sequencing justifies skipping RAPD enrichment of circRNAs as done by the authors, I don’t know.
While the findings of the paper are relevant and novel, the methodology used is flawed.
Author Response
Reviewer 1
- The introduction is well-written and on point.
Our response: We thank the reviewer.
- The Results section is way too detailed and can be modified to move details about protocols or findings to the Methods or Discussions section, respectively. For example, see lines: 171-175, 204-213, 252-256, 303-309, etc. This, I think, could increase the readability of the manuscript.
Many of the findings in the Results section can be summarised in tabular or picture format instead and the implications of these findings could be discussed in the Discussion section. For example, see subsection 3.3.
Line 84: Is it ‘heat-activated’ or ‘heat-inactivated’?
- Line 90: remove ‘of’.
- Line 493: remove ‘NET-seq peaks’
- Lines 212-215: It should be Fig. S2 instead of Fig. 2.
- Lines 247-248: Sentence not clear.
- The western blot given in Fig. S1 should be published as part of the main manuscript, not in the supplementary files.
- Original pictures of gels in the supplementary files have not been uploaded.
Our response: We thank the reviewer. We have summarized the results and have corrected the above mistakes. We have also added a table to show the numbers drawn in the figures for easy reading.
Major issues:
- The western blot in the supplementary file doesn’t match the original picture of the image provided. Please check.
Our response: We apologize and have now corrected.
- The numbers in the Venn diagrams (Fig. 1C, D) don’t add up. Please check. Fig. 1B, what does row 1 represent?
Our response: We apologize and have now corrected.
- The marker lane in the WB seems to be pasted from another blot.
Our response: The marker is from a Coomassie staining and aligned to the western blot. We now state so in the legend and make a box surrounding that for clarity.
- The most important issue with the paper is that the authors have sequenced the total RNA, instead of going for the enrichment of the important fraction (i.e., circRNA).
Our response: We thank the reviewer, and we have now corrected our mistake. In fact, we performed ribosomal depleted RNA-seq not total RNA. We apologize and have now corrected. For the RNAseq experiment total RNA underwent ribosomal RNA-depletion, and each sample was read at a depth of >100 million reads to detect circular RNA (Table provided below).
- Since circRNAs occur in a very low proportion, I don’t think the results are reliable enough in the absence of RNAse R digestion or RAPD. This is complicated by the fact that the authors have centered all of their analysis around the circRNA fraction while sequencing total RNA and skipping almost the entire part about mRNAs.
Our method of analysis, SEEKCrit, is designed to measure circRNA from total or ribosomal depleted RNA. It measures back splice junctions in relation to adjacent linear junctions to determine how back splicing changes in relation to linear splicing of the same gene. identifies the most prevalent circRNAs, which was the goal here. Papers comparing this to linear digestion include (Cheng, et al. 2015; López-Jiménez, et al. 2018; Sekar, et al. 2019; Zhang, et al. 2014). Thus, while linear digestion may enrich circular RNAs, it also digests some of them as well, and deep ribosomal depleted sequencing data has been used for identification. Lastly, we already had performed PARP1’s role in exon skipping in our previous publication (Matveeva, et al. 2016a).
- Also, since I was not able to access the raw FASTQ files and the authors have not mentioned the sequence depth/coverage, I cannot appreciate how truly the findings represent the actual cellular/molecular picture. A table showing how much of the sequence data aligned with each RNA fraction (mRNA, circRNA, other non-coding RNA types) would have been of immense help. Whether the depth of sequencing justifies skipping RAPD enrichment of circRNAs as done by the authors, I don’t know.
Our response: We thank the reviewer. Below are the read numbers, the aligned pairs. We also show the chimeric reads and percentages. Chimeric reads are indicative of either circular RNAs or gene fusion events. We therefore think that read numbers suffice for detecting circRNAs.
|
File |
Input Reads |
Aligned Pairs |
Concordant Pair Alignment Rate |
Chimeric junction reads |
Percent of reads with chimeric junctions (%) |
|
WT – replicate 1 |
112,548,973 |
96,764,230 |
85.97% |
454964 |
0.404 |
|
WT – replicate 2 |
118,443,341 |
103,062,882 |
87.01% |
463967 |
0.392 |
|
WT – replicate 3 |
116,256,543 |
101,566,816 |
87.36% |
442150 |
0.380 |
|
PARPi - replicate 1 |
111,561,196 |
97,716,946 |
87.59% |
441903 |
0.396 |
|
PARPi - replicate 2 |
107,707,449 |
93,700,340 |
86.99% |
422025 |
0.392 |
|
PARPi - replicate 3 |
117,113,516 |
101,231,327 |
86.43% |
425180 |
0.393 |
|
KD – replicate 1 |
116,003,671 |
99,674,367 |
85.92% |
444543 |
0.383 |
|
KD – replicate 2 |
103,827,822 |
88,616,518 |
85.34% |
401996 |
0.387 |
|
KD – replicate 3 |
109,774,312 |
94,387,504 |
85.98% |
420897 |
0.383 |
- While the findings of the paper are relevant and novel, the methodology used is flawed.
Our response: We are grateful to the reviewer but also beg to differ. Several studies have shown that circRNAs can be detected from ribo-depleted RNA-seq (Cheng, et al. 2015; López-Jiménez, et al. 2018; Sekar, et al. 2019; Zhang, et al. 2014). We are highlighting a few. In addition. there is no perfect method for detecting circRNAs (outlined clearly in (Szabo and Salzman 2016; Vromman, et al. 2020) We used several stringent criteria to validate our detected circRNAs and further analyses, used only the stringently detected circRNAs. As techniques develop to work with these types of RNAs, we also hope our data will provide a platform to begin to tease some of the mechanistic details that might be missing.
References:
Cheng, Jun, Franziska Metge, and Christoph Dieterich
2015 Specific identification and quantification of circular RNAs from sequencing data. Bioinformatics 32(7):1094-1096.
Dhahri, H., E. Matveeva, and Y. Fondufe-Mittendorf
2023 Approach to Measuring the Effect of PARP1 on RNA Polymerase II Elongation Rates. Methods Mol Biol 2609:315-328.
Erener, S., et al.
2012 Poly(ADP-ribose)polymerase-1 (PARP1) controls adipogenic gene expression and adipocyte function. Mol Endocrinol 26(1):79-86.
Frizzell, K. M., et al.
2009 Global analysis of transcriptional regulation by poly(ADP-ribose) polymerase-1 and poly(ADP-ribose) glycohydrolase in MCF-7 human breast cancer cells. J Biol Chem 284(49):33926-38.
Gibson, B. A., and W. L. Kraus
2012 New insights into the molecular and cellular functions of poly(ADP-ribose) and PARPs. Nat Rev Mol Cell Biol 13(7):411-24.
Gibson, B. A., et al.
2016 Chemical genetic discovery of PARP targets reveals a role for PARP-1 in transcription elongation. Science 353(6294):45-50.
Huang, D., and W. L. Kraus
2022 The expanding universe of PARP1-mediated molecular and therapeutic mechanisms. Mol Cell.
Krishnakumar, R., et al.
2008 Reciprocal binding of PARP-1 and histone H1 at promoters specifies transcriptional outcomes. Science 319(5864):819-21.
Krishnakumar, R., and W. L. Kraus
2010 PARP-1 regulates chromatin structure and transcription through a KDM5B-dependent pathway. Mol Cell 39(5):736-49.
Kristensen, L. S.
2021 Profiling of circRNAs using an enzyme-free digital counting method. Methods 196:11-16.
Leutert, M., D. M. L. Pedrioli, and M. O. Hottiger
2016 Identification of PARP-Specific ADP-Ribosylation Targets Reveals a Regulatory Function for ADP-Ribosylation in Transcription Elongation. Mol Cell 63(2):181-183.
López-Jiménez, E., A. M. Rojas, and E. Andrés-León
2018 RNA sequencing and Prediction Tools for Circular RNAs Analysis. Adv Exp Med Biol 1087:17-33.
Matveeva, E. A., et al.
2019a Coupling of PARP1-mediated chromatin structural changes to transcriptional RNA polymerase II elongation and cotranscriptional splicing. Epigenetics Chromatin 12(1):15.
Matveeva, E. A., H. Dhahri, and Y. Fondufe-Mittendorf
2022 PARP1's Involvement in RNA Polymerase II Elongation: Pausing and Releasing Regulation through the Integrator and Super Elongation Complex. Cells 11(20).
Matveeva, E. A., L. F. Mathbout, and Y. N. Fondufe-Mittendorf
2019b PARP1 is a versatile factor in the regulation of mRNA stability and decay. Sci Rep 9(1):3722.
Matveeva, E., et al.
2016a Involvement of PARP1 in the regulation of alternative splicing. Cell Discov 2:15046.
Matveeva, Elena A., et al.
2019c Coupling of PARP1-mediated chromatin structural changes to transcriptional RNA polymerase II elongation and cotranscriptional splicing. Epigenetics & Chromatin 12(1):15.
Matveeva, Elena, et al.
2016b Involvement of PARP1 in the regulation of alternative splicing. Cell Discovery 2(1):15046.
Melikishvili, M., E. Matveeva, and Y. Fondufe-Mittendorf
2017 Methodology to Identify Poly-ADP-Ribose Polymerase 1 (PARP1)-mRNA Targets by PAR-CLiP. Methods Mol Biol 1608:211-228.
Nalabothula, N., et al.
2015 Genome-Wide Profiling of PARP1 Reveals an Interplay with Gene Regulatory Regions and DNA Methylation. PLoS One 10(8):e0135410.
Petesch, S. J., and J. T. Lis
2008 Rapid, transcription-independent loss of nucleosomes over a large chromatin domain at Hsp70 loci. Cell 134(1):74-84.
—
2012a Activator-induced spread of poly(ADP-ribose) polymerase promotes nucleosome loss at Hsp70. Mol Cell 45(1):64-74.
—
2012b Overcoming the nucleosome barrier during transcript elongation. Trends Genet 28(6):285-94.
Puentes-Pardo, J. D., et al.
2023 PARP-1 Expression Influences Cancer Stem Cell Phenotype in Colorectal Cancer Depending on p53. Int J Mol Sci 24(5).
Sekar, S., et al.
2019 Identification of Circular RNAs using RNA Sequencing. J Vis Exp (153).
Shan, L., et al.
2014 GATA3 cooperates with PARP1 to regulate CCND1 transcription through modulating histone H1 incorporation. Oncogene 33(24):3205-16.
Siraj, A. K., et al.
2018 Overexpression of PARP is an independent prognostic marker for poor survival in Middle Eastern breast cancer and its inhibition can be enhanced with embelin co-treatment. Oncotarget 9(99):37319-37332.
Szabo, L., and J. Salzman
2016 Detecting circular RNAs: bioinformatic and experimental challenges. Nat Rev Genet 17(11):679-692.
Vromman, Marieke, Jo Vandesompele, and Pieter-Jan Volders
2020 Closing the circle: current state and perspectives of circular RNA databases. Briefings in Bioinformatics 22(1):288-297.
Zhang, Z., et al.
2014 Discovery of replicating circular RNAs by RNA-seq and computational algorithms. PLoS Pathog 10(12):e1004553.
Zhao, H., et al.
2015 PARP1- and CTCF-Mediated Interactions between Active and Repressed Chromatin at the Lamina Promote Oscillating Transcription. Mol Cell 59(6):984-97.
Reviewer 2 Report
The manuscript of Eleazer et al describes the role of PARP1 in circular RNA biogenesis.
This work is potentially interesting, more clear in the first part of the results rather than the last.
Major criticisms
- The entire work has been conducted using only one cell line. Additional cell lines sholud be necessary to strength the results.
- To support the idea PARP1 mediates gene expression than PARylation, additional evidence should be provided (i.e. genome wide occupancy of PARP1, ectopic overexpression of specific mutants.
- Additional evidence of RNAPII pausing after parp kd should be provided (i.e. analysis of active forms –ser2p or ser5p- of RNAPII).
- It would be very interesting to evaluate the effects of PARPi in NET-seq.
Minor comments:
- There is not correspondence between supplementary Figure 2a and 2b and the main text.
- It is not clear what is AS in figure 3b-c.
- Result of chapter 3.3 should be re-written according to results section.
Author Response
Reviewer 2
Major criticisms
- The entire work has been conducted using only one cell line. Additional cell lines should be necessary to strength the results.
Our response: This is a great idea, which we will hope to recapitulate in future studies. In almost all studies understanding PARP1 in chromatin biology and gene regulation, only one cell line has been used.
- The role of PARP1 in nucleosome disruption at promoters using S2 Drosophila cells just like us (Petesch and Lis 2008; Petesch and Lis 2012a; Petesch and Lis 2012b)
- The role of PARP1 in chromatin modulation in MCF7 cells (Frizzell, et al. 2009; Gibson and Kraus 2012; Krishnakumar, et al. 2008; Krishnakumar and Kraus 2010)
- Genome-wide mapping of PARP1 binding in S2 cells (Nalabothula, et al. 2015)
- PARP1 and CTCF mediation in HCT116 cells (Zhao, et al. 2015)
- PARP1 in adipocyte function using 3T3-L1 cell line (Erener, et al. 2012)
- PARP1 modulating GATA-3 using MCF7 cells – (Shan, et al. 2014)
- PARP1 in alternative splicing in S2 drosophila cells (Matveeva, et al. 2019a; Matveeva, et al. 2022; Matveeva, et al. 2019b; Matveeva, et al. 2016a);
- Gene-specific PARP1 chromatin binding in MCF cells (Frizzell, et al. 2009; Gibson, et al. 2016; Krishnakumar, et al. 2008; Krishnakumar and Kraus 2010).
We highlight only a few studies with key discoveries made in only one cell line. These studies and many more, all provided critical understanding of PARP1 biology using only a single cell line which was then recapitulated in other studies.
- To support the idea PARP1 mediates gene expression than PARylation, additional evidence should be provided (i.e., genome wide occupancy of PARP1, ectopic overexpression of specific mutants.
Our response: We already had presented studies previously on PARP1’s role in gene expression. We also had shown its genome-wide occupancy (Nalabothula, et al. 2015); its role on alternative splicing (Matveeva, et al. 2019a; Matveeva, et al. 2022; Matveeva, et al. 2019b; Matveeva, et al. 2016a; Melikishvili, et al. 2017). Thus PARP1’s role in gene expression is well-established by us and others (Frizzell, et al. 2009; Gibson, et al. 2016; Huang and Kraus 2022; Krishnakumar, et al. 2008; Krishnakumar and Kraus 2010). We do not think mutating PARP1 will add to the story. If we mutate the N-terminus with the zinc-finger binding, domain then PARP1 may or may no longer bind to DNA. 2. If we mutate the catalytic site, then PARP1 is no longer able to activate splicing factors. In both cases, we are not measuring the normal effect of PARP1 in the cell, rather an artificial system, which is not relevant to the study.
- Additional evidence of RNAPII pausing after parp kd should be provided (i.e., analysis of active forms –ser2p or ser5p- of RNAPII).
Our response: We had already shown in our previous studies and which we highlight in the manuscript, PARP1’s occupancy, in relation to total RNAPII, Ser2p, Ser5p and the other forms of RNAPII along genes in the presence and absence of PARP1 or its catalytic activity (Matveeva et al, 2019; Matveeva et al, 2023).
- It would be very interesting to evaluate the effects of PARPi in NET-seq.
Our response: We agree that examining the effects of PARylation on RNAPII pausing would be an excellent contribution to the field. We plan on examining this specific effect of PARP1 on elongation and back splicing in the future.
- Minor comments:
- There is not correspondence between supplementary Figure 2a and 2b and the main text.
- It is not clear what is AS in figure 3b-c.
- Result of chapter 3.3 should be re-written according to results section.
Our response: We thank the reviewer and have corrected all the above.
References:
Cheng, Jun, Franziska Metge, and Christoph Dieterich
2015 Specific identification and quantification of circular RNAs from sequencing data. Bioinformatics 32(7):1094-1096.
Dhahri, H., E. Matveeva, and Y. Fondufe-Mittendorf
2023 Approach to Measuring the Effect of PARP1 on RNA Polymerase II Elongation Rates. Methods Mol Biol 2609:315-328.
Erener, S., et al.
2012 Poly(ADP-ribose)polymerase-1 (PARP1) controls adipogenic gene expression and adipocyte function. Mol Endocrinol 26(1):79-86.
Frizzell, K. M., et al.
2009 Global analysis of transcriptional regulation by poly(ADP-ribose) polymerase-1 and poly(ADP-ribose) glycohydrolase in MCF-7 human breast cancer cells. J Biol Chem 284(49):33926-38.
Gibson, B. A., and W. L. Kraus
2012 New insights into the molecular and cellular functions of poly(ADP-ribose) and PARPs. Nat Rev Mol Cell Biol 13(7):411-24.
Gibson, B. A., et al.
2016 Chemical genetic discovery of PARP targets reveals a role for PARP-1 in transcription elongation. Science 353(6294):45-50.
Huang, D., and W. L. Kraus
2022 The expanding universe of PARP1-mediated molecular and therapeutic mechanisms. Mol Cell.
Krishnakumar, R., et al.
2008 Reciprocal binding of PARP-1 and histone H1 at promoters specifies transcriptional outcomes. Science 319(5864):819-21.
Krishnakumar, R., and W. L. Kraus
2010 PARP-1 regulates chromatin structure and transcription through a KDM5B-dependent pathway. Mol Cell 39(5):736-49.
Kristensen, L. S.
2021 Profiling of circRNAs using an enzyme-free digital counting method. Methods 196:11-16.
Leutert, M., D. M. L. Pedrioli, and M. O. Hottiger
2016 Identification of PARP-Specific ADP-Ribosylation Targets Reveals a Regulatory Function for ADP-Ribosylation in Transcription Elongation. Mol Cell 63(2):181-183.
López-Jiménez, E., A. M. Rojas, and E. Andrés-León
2018 RNA sequencing and Prediction Tools for Circular RNAs Analysis. Adv Exp Med Biol 1087:17-33.
Matveeva, E. A., et al.
2019a Coupling of PARP1-mediated chromatin structural changes to transcriptional RNA polymerase II elongation and cotranscriptional splicing. Epigenetics Chromatin 12(1):15.
Matveeva, E. A., H. Dhahri, and Y. Fondufe-Mittendorf
2022 PARP1's Involvement in RNA Polymerase II Elongation: Pausing and Releasing Regulation through the Integrator and Super Elongation Complex. Cells 11(20).
Matveeva, E. A., L. F. Mathbout, and Y. N. Fondufe-Mittendorf
2019b PARP1 is a versatile factor in the regulation of mRNA stability and decay. Sci Rep 9(1):3722.
Matveeva, E., et al.
2016a Involvement of PARP1 in the regulation of alternative splicing. Cell Discov 2:15046.
Matveeva, Elena A., et al.
2019c Coupling of PARP1-mediated chromatin structural changes to transcriptional RNA polymerase II elongation and cotranscriptional splicing. Epigenetics & Chromatin 12(1):15.
Matveeva, Elena, et al.
2016b Involvement of PARP1 in the regulation of alternative splicing. Cell Discovery 2(1):15046.
Melikishvili, M., E. Matveeva, and Y. Fondufe-Mittendorf
2017 Methodology to Identify Poly-ADP-Ribose Polymerase 1 (PARP1)-mRNA Targets by PAR-CLiP. Methods Mol Biol 1608:211-228.
Nalabothula, N., et al.
2015 Genome-Wide Profiling of PARP1 Reveals an Interplay with Gene Regulatory Regions and DNA Methylation. PLoS One 10(8):e0135410.
Petesch, S. J., and J. T. Lis
2008 Rapid, transcription-independent loss of nucleosomes over a large chromatin domain at Hsp70 loci. Cell 134(1):74-84.
—
2012a Activator-induced spread of poly(ADP-ribose) polymerase promotes nucleosome loss at Hsp70. Mol Cell 45(1):64-74.
—
2012b Overcoming the nucleosome barrier during transcript elongation. Trends Genet 28(6):285-94.
Puentes-Pardo, J. D., et al.
2023 PARP-1 Expression Influences Cancer Stem Cell Phenotype in Colorectal Cancer Depending on p53. Int J Mol Sci 24(5).
Sekar, S., et al.
2019 Identification of Circular RNAs using RNA Sequencing. J Vis Exp (153).
Shan, L., et al.
2014 GATA3 cooperates with PARP1 to regulate CCND1 transcription through modulating histone H1 incorporation. Oncogene 33(24):3205-16.
Siraj, A. K., et al.
2018 Overexpression of PARP is an independent prognostic marker for poor survival in Middle Eastern breast cancer and its inhibition can be enhanced with embelin co-treatment. Oncotarget 9(99):37319-37332.
Szabo, L., and J. Salzman
2016 Detecting circular RNAs: bioinformatic and experimental challenges. Nat Rev Genet 17(11):679-692.
Vromman, Marieke, Jo Vandesompele, and Pieter-Jan Volders
2020 Closing the circle: current state and perspectives of circular RNA databases. Briefings in Bioinformatics 22(1):288-297.
Zhang, Z., et al.
2014 Discovery of replicating circular RNAs by RNA-seq and computational algorithms. PLoS Pathog 10(12):e1004553.
Zhao, H., et al.
2015 PARP1- and CTCF-Mediated Interactions between Active and Repressed Chromatin at the Lamina Promote Oscillating Transcription. Mol Cell 59(6):984-97.
Reviewer 3 Report
In this manuscript, the authors conducted bioinformatics analysis and found the possible role of PARP1 in regulating circRNA biogenesis. The authors identified many unique circRNAs in PARP1 knocking down and PARP1i cells. Finally, the authors found a difference in regulating RNAPII pausing mediated by PARP1. In general, my assessment of this manuscript is positive, although I believe that it needs to be further improved prior to publication in Cells.
Specific comments:
1. In this manuscript, the authors conducted analysis in the PARP1 knocking down and PARP1i cells, my concern is how about in the PARP1 overexpression cells?
2. PARP1 is an important DNA repair protein. Does PARP1 affect the circRNAs that regulate the DNA damage and repair related gene expression?
3. In figure S1, the authors should add the molecular weight of marker to indicate the MW of PARP1 and actin.
Author Response
Reviewer 3:
In this manuscript, the authors conducted bioinformatics analysis and found the possible role of PARP1 in regulating circRNA biogenesis. The authors identified many unique circRNAs in PARP1 knocking down and PARP1i cells. Finally, the authors found a difference in regulating RNAPII pausing mediated by PARP1. In general, my assessment of this manuscript is positive, although I believe that it needs to be further improved prior to publication in Cells.
Specific comments:
- In this manuscript, the authors conducted analysis in the PARP1 knocking down and PARP1i cells, my concern is how about in the PARP1 overexpression cells?
Our response: We thank the reviewer for bringing this up. We however think this is not a logical experiment to perform. This is because we seek to understand PARP1’s role in normal growth conditions. And even in these normal conditions, PARP1 is abundantly expressed. In cancer conditions, PARP1 is expressed even more. (Puentes-Pardo, et al. 2023; Siraj, et al. 2018). Overexpression studies as proposed, would likely lead to spurious incorporation of PARP1 into regions of the genome that are not biologically relevant to our inquiry. As we are interested in the normal functioning of PARP1, and not its pathological role in human disease, overexpression of PARP1 within our model would not likely provide data aligned with our line of inquiry.
- PARP1 is an important DNA repair protein. Does PARP1 affect the circRNAs that regulate the DNA damage and repair related gene expression?
Our response: This is a fantastic idea, and it will be great to know but the functions of many circRNAs remains unknown. Future studies will be needed to see if any of these circRNAs we identified play a role in DNA repair.
- In figure S1, the authors should add the molecular weight of marker to indicate the MW of PARP1 and actin.
Our response: Thank you. We have included this now.
References:
Cheng, Jun, Franziska Metge, and Christoph Dieterich
2015 Specific identification and quantification of circular RNAs from sequencing data. Bioinformatics 32(7):1094-1096.
Dhahri, H., E. Matveeva, and Y. Fondufe-Mittendorf
2023 Approach to Measuring the Effect of PARP1 on RNA Polymerase II Elongation Rates. Methods Mol Biol 2609:315-328.
Erener, S., et al.
2012 Poly(ADP-ribose)polymerase-1 (PARP1) controls adipogenic gene expression and adipocyte function. Mol Endocrinol 26(1):79-86.
Frizzell, K. M., et al.
2009 Global analysis of transcriptional regulation by poly(ADP-ribose) polymerase-1 and poly(ADP-ribose) glycohydrolase in MCF-7 human breast cancer cells. J Biol Chem 284(49):33926-38.
Gibson, B. A., and W. L. Kraus
2012 New insights into the molecular and cellular functions of poly(ADP-ribose) and PARPs. Nat Rev Mol Cell Biol 13(7):411-24.
Gibson, B. A., et al.
2016 Chemical genetic discovery of PARP targets reveals a role for PARP-1 in transcription elongation. Science 353(6294):45-50.
Huang, D., and W. L. Kraus
2022 The expanding universe of PARP1-mediated molecular and therapeutic mechanisms. Mol Cell.
Krishnakumar, R., et al.
2008 Reciprocal binding of PARP-1 and histone H1 at promoters specifies transcriptional outcomes. Science 319(5864):819-21.
Krishnakumar, R., and W. L. Kraus
2010 PARP-1 regulates chromatin structure and transcription through a KDM5B-dependent pathway. Mol Cell 39(5):736-49.
Kristensen, L. S.
2021 Profiling of circRNAs using an enzyme-free digital counting method. Methods 196:11-16.
Leutert, M., D. M. L. Pedrioli, and M. O. Hottiger
2016 Identification of PARP-Specific ADP-Ribosylation Targets Reveals a Regulatory Function for ADP-Ribosylation in Transcription Elongation. Mol Cell 63(2):181-183.
López-Jiménez, E., A. M. Rojas, and E. Andrés-León
2018 RNA sequencing and Prediction Tools for Circular RNAs Analysis. Adv Exp Med Biol 1087:17-33.
Matveeva, E. A., et al.
2019a Coupling of PARP1-mediated chromatin structural changes to transcriptional RNA polymerase II elongation and cotranscriptional splicing. Epigenetics Chromatin 12(1):15.
Matveeva, E. A., H. Dhahri, and Y. Fondufe-Mittendorf
2022 PARP1's Involvement in RNA Polymerase II Elongation: Pausing and Releasing Regulation through the Integrator and Super Elongation Complex. Cells 11(20).
Matveeva, E. A., L. F. Mathbout, and Y. N. Fondufe-Mittendorf
2019b PARP1 is a versatile factor in the regulation of mRNA stability and decay. Sci Rep 9(1):3722.
Matveeva, E., et al.
2016a Involvement of PARP1 in the regulation of alternative splicing. Cell Discov 2:15046.
Matveeva, Elena A., et al.
2019c Coupling of PARP1-mediated chromatin structural changes to transcriptional RNA polymerase II elongation and cotranscriptional splicing. Epigenetics & Chromatin 12(1):15.
Matveeva, Elena, et al.
2016b Involvement of PARP1 in the regulation of alternative splicing. Cell Discovery 2(1):15046.
Melikishvili, M., E. Matveeva, and Y. Fondufe-Mittendorf
2017 Methodology to Identify Poly-ADP-Ribose Polymerase 1 (PARP1)-mRNA Targets by PAR-CLiP. Methods Mol Biol 1608:211-228.
Nalabothula, N., et al.
2015 Genome-Wide Profiling of PARP1 Reveals an Interplay with Gene Regulatory Regions and DNA Methylation. PLoS One 10(8):e0135410.
Petesch, S. J., and J. T. Lis
2008 Rapid, transcription-independent loss of nucleosomes over a large chromatin domain at Hsp70 loci. Cell 134(1):74-84.
—
2012a Activator-induced spread of poly(ADP-ribose) polymerase promotes nucleosome loss at Hsp70. Mol Cell 45(1):64-74.
—
2012b Overcoming the nucleosome barrier during transcript elongation. Trends Genet 28(6):285-94.
Puentes-Pardo, J. D., et al.
2023 PARP-1 Expression Influences Cancer Stem Cell Phenotype in Colorectal Cancer Depending on p53. Int J Mol Sci 24(5).
Sekar, S., et al.
2019 Identification of Circular RNAs using RNA Sequencing. J Vis Exp (153).
Shan, L., et al.
2014 GATA3 cooperates with PARP1 to regulate CCND1 transcription through modulating histone H1 incorporation. Oncogene 33(24):3205-16.
Siraj, A. K., et al.
2018 Overexpression of PARP is an independent prognostic marker for poor survival in Middle Eastern breast cancer and its inhibition can be enhanced with embelin co-treatment. Oncotarget 9(99):37319-37332.
Szabo, L., and J. Salzman
2016 Detecting circular RNAs: bioinformatic and experimental challenges. Nat Rev Genet 17(11):679-692.
Vromman, Marieke, Jo Vandesompele, and Pieter-Jan Volders
2020 Closing the circle: current state and perspectives of circular RNA databases. Briefings in Bioinformatics 22(1):288-297.
Zhang, Z., et al.
2014 Discovery of replicating circular RNAs by RNA-seq and computational algorithms. PLoS Pathog 10(12):e1004553.
Zhao, H., et al.
2015 PARP1- and CTCF-Mediated Interactions between Active and Repressed Chromatin at the Lamina Promote Oscillating Transcription. Mol Cell 59(6):984-97.
Round 2
Reviewer 1 Report
Thanks to the authors for detailed responses to all my comments from the previous round.
The only reason why I commented during round 1 that the metholody was flawed is because at that time the authors had not mentioned the ribo-depletion step before going for RNA-Seq. Now that the authors have mentioned that it was actually done, I agree with the explanation provided after that and believe the manuscript makes perfect sense. Also, other issues have been addressed. One more suggestion: the table provided in response to one of my comments above can be added in supplementary data, if that has not been done already.
Reviewer 2 Report
Dear Authors,
my criticisms have not been addressed convincingly.
Reviewer 3 Report
accept